# Experimental Study on the Efficacy of a Novel Personal Cooling Vest Incorporated with Phase Change Materials and Fans

**DOI:** 10.3390/ma13081801

**Published:** 2020-04-11

**Authors:** Xiaoyang Ni, Tianyu Yao, Ying Zhang, Yijie Zhao, Qin Hu, Albert P.C. Chan

**Affiliations:** 1Faculty of Engineering, China University of Geosciences (Wuhan), Wuhan 430074, China; xy_ni@163.com (X.N.); samuelying@foxmail.com (T.Y.); 2School of Safety Science and Emergency Management, Wuhan University of Technology, Wuhan 430070, China; qinqin@whut.edu.cn; 3Department of Building and Real Estate, The Hong Kong Polytechnic University, Hung Hom, Hong Kong, China; zyj336699@126.com

**Keywords:** personal cooling, phase change material, heat strain, thermal load, cumulative heat storage

## Abstract

In recent years, personal cooling has aroused much attention because it can achieve both localized high-level thermal comfort and build energy savings. In this study, a novel hybrid personal cooling vest (PCV) incorporated with phase change materials (PCMs) and ventilation fans was developed, and its efficacy was investigated by human trials in a hot-humid climate chamber. Three generally accepted indices (thermal load, *Q*; thermal sensation, TS; and physiological strain index, PSI) and a new proposed index (cumulative heat storage, CHS) during human trials were comparatively studied between the two human trial groups, i.e., the PCV group (wearing the PCV) and the CON group (without PCV). Results found that TS, PSI, and CHS were significantly reduced by the PCV, which suggests that the PCV can significantly improve both the perceptual and physiological strain. In addition, a strong linear relationship (*r*^2^ = 0.8407) was found between the proposed index of CHS with PSI, which indicates the applicability and reliability of CHS for assessing physiological heat strain.

## 1. Introduction

Construction workers in Hong Kong are particularly under the high risk of heat stress and heat related accidents in the hot and humid summer [1,2]. The chance of risky behaviors increased concomitantly with environmental temperatures above 24 °C [3]. Personal cooling is one of the methods suggested for decreasing heat stress and improving thermal comfort of outdoor workers; since, in such conditions, air-conditioning systems are not effective [4,5,6,7].

Current personal cooling systems are of primarily three types, these include air cooling systems, liquid cooling systems, and phase change cooling systems. Though the air cooling system or liquid cooling system can provide enough cooling power for the construction workers, they restrict the worker’s movement and bring inconveniences due to the large amount of weight of the compression systems and pumps [8]. Thus, phase change cooling system aroused much attention because it does not need additional cooling devices but employs the phase change material to absorb a substantial amount of latent heat [9,10,11]. Though the phase change cooling systems are more portable, its cooling capacity is generally limited, and it needs other cooling devices.

The cooling performance of personal cooling systems are generally experimentally investigated by two main methods: manikin testing and human trials [12]. Lots of research was carried out through manikin testing for its lower cost compared to human trials [13,14]. However, manikins showed some technical limitations in combining the thermo-physiological model with a thermal manikin and are often unsatisfactory for validation [15,16]. Human trials are the most direct and appropriate way for the assessment of the cooling performance of personal cooling systems, even though it is expensive and time-consuming [17].

In this paper, we developed a hybrid cooling vest for construction workers in Hong Kong and investigated its cooling performance with human trials in a climate chamber. The physiological parameters, including the core temperature, the skin temperature, and the heart rate, were measured. Two widely accepted heat strain indices, thermal sensation [18,19,20] and physiological strain index (PSI) [21,22,23], were employed to assess the cooling performance from the aspect of perceptual and physiological strain, respectively. In addition, a body temperature based heat strain index of cumulative heat storage (CHS) was proposed, and its applicability for heat strain assessment was examined. If approved, heat strain could be monitored by infrared thermography (IRT) rather than some other invasive diagnostic tool [24,25], such as the ingestible capsule or a temperature sensor inserted into the rectum, which are very uncomfortable or high-cost.

## 2. Mathematical Models

The human thermal load, *Q*, is in itself the heat flow rate in a given condition and calculated from Equation (1) as the remaining amount of each item of energy balance [26,27]:*Q* = *M* − *W* + *R*_net_ − *E* − *C*(1)
where *M* is the metabolism, *W* is the workload, *R*_net_ is the net radiation, *E* is the latent heat loss, and *C* is the sensible heat loss. Sensible heat loss (*C*) was the sum of convective heat loss from skin (*C*_sk_) and expiration (*C*_res_), namely *C* = *C*_sk_ + *C*_res_. The latent heat loss, *E*, is the sum of evaporated heat from skin (*E*_sk_) and breathing (*E*_res_), namely *E* = *E*_sk_ + *E*_res_ [27]. The heat balance can also be expressed as Equation (2), based on the dynamic two-node model [28]:*M* − *W* = *Q*_sk_ + *Q*_res_ + *S* = *C*_sk_ + *C*_res_ + *R* + *E*_sk_ + *E*_res_ + *S*_sk_ + *S*_c_(2)
The net heat production *M* − *W* is either dissipated through the skin surface (*Q*_sk_ = *C*_sk_ + *E*_sk_) and respiratory tract (*Q*_res_ = *C*_res_ + *E*_res_), or stored (*S* = *S*_sk_ + *S*_c_). *S*_sk_ and *S*_c_ are given by:(3)$$S_{\mathrm{sk}}=\frac{\alpha c_{p}m}{A_{D}}\frac{dT_{\mathrm{sk}}}{dt}$$
(4)$$S_{c}=\frac{\left(1-\alpha \right)c_{p}m}{A_{D}}\frac{dT_{c}}{dt}$$
where *c*_p_ is the specific heat; *m* and *A*_D_ are the mass and area of the human body, respectively; and *T*_sk_ and *T*_c_ are the temperatures of the skin and core body, respectively. The fraction skin mass, α, depends on the rate of blood flowing to the skin surface, and is given by α = 0.3 − 0.9(*T*_c_ − 36.8), and limited to 0.1 for *T*_c_ > 39 °C and 0.3 for *T*_c_ < 36.8 °C [18].

Combining Equations (3) and (4), thermal load can be obtained by the sum of the rates of heat storage in the skin compartment and core compartment, i.e.:(5)$$Q=S_{\mathrm{sk}}+S_{c}=\frac{c_{p}m}{A_{D}}\left[\alpha \frac{dT_{\mathrm{sk}}}{dt}+\left(1-\alpha \right)\frac{dT_{c}}{dt}\right]$$

The cumulative heat storage in the human body (CHS) per area is calculated using the following equation:(6)$$CHS=\int_{0}^{t}Qd\tau$$

By substituting Equation (5) into Equation (6) and estimating d*T*/d*t* with Δ*T*/Δ*t*, CHS is rewritten as:(7)$$CHS=\frac{c_{p}m}{A_{D}}\left[\alpha (T_{\mathrm{sk}}-T_{\mathrm{sk},0})+\left(1-\alpha \right)(T_{c}-T_{c,0})\right]$$
where *T*_sk,0_ and *T*_c,0_ are the initial temperatures of the skin and core compartments, respectively.

## 3. Experiments

### 3.1. Experimental Design

The personal cooling vest (PCV) is a newly developed vest for construction workers, with a total weight of 1.26 kg, as illustrated in Figure 1. It incorporated eight packs of phase change materials (PCMs) (Climator Sweden AB, Skövde, Sweden) and two ventilation fans. PCMs were placed, with two packs in the chest region, two packs in the abdomen region, and four packs in the back. The phase change temperature of PCMs is 28 °C, and its latent heat is 131 kJ/kg. The ventilation fans were placed at the lower back region, as shown in Figure 1b. More information about this PCV can be found in the previous publication [2].

The experiments were carried out in a climate chamber (3 m × 2.5 m × 2.2 m), in which the air temperature and relative humidity could be controlled. In this study, the temperature and relative humidity in the climate chamber were maintained at 37 °C and 60%, respectively, which was similar to the stressful conditions of construction sites in Hong Kong [19,20]. A total of 12 healthy males were enrolled in this study, and their mean age, height, and weight were 22.08 ± 3.32, 1.70 ± 0.05 m, and 61.08 ± 8.05 kg, respectively. All the subjects were informed of the protocol of the experiment, in which the aim, details, and potential medical risks with this study were fully explained. Subjects were asked to sign the consent form if they agreed to be a part of the study. They were also notified that they could quit this study at any time without penalty. The research was conducted with the approval of the Hong Kong Polytechnic University’s Human Subjects Ethics Sub-committee.

The experimental protocol is illustrated in Figure 2, which includes five stages: (1) 30 min of pre-exercise rest (rest), (2) the first period of intermittent running (1st run), (3) 6 min of active recovery, (4) 30 min of passive recovery, and (5) the second period of intermittent running (2nd run). During the periods of 1st run and 2nd run, subjects were required to run on the treadmill with a certain slope and speed to mimic construction work [29]. Each subject was required to undertake two wear trials, in one of which the subjects wore a personal cooling vest (PCV) during the stage of passive recovery as the cooling group, and the other without PCV as the control group (CON).

Subjects reported to the laboratory at the same time of day, spaced at least one week apart. They were required not to drink alcohol at least 24 h prior to each test. Upon arrival, they rested in a seated position for nearly 30 min. During the rest, they were briefed on the experiment, signed the consent form, and completed the basic demographic information, such as name, height, and age. Subjects then drank a cup of warm water at 37 °C (3 mL water per kg body weight). After that, they took off their own clothes and weighed for body mass (with underwear). The subjects were instructed to put on their assigned construction uniform and were equipped with sensors for measuring heart rate, skin, and core temperatures.

After the above preparation, the subject entered the climate chamber with designed climate condition. Firstly, the participant took a pre-exercise rest on a backless seat for 30 min, and then performed intermittent running (1st run), as shown in Figure 2, on a motorized treadmill (pulsar 4.0, h/p/cosmos, Traunstein, Germany). The run lasted no more than 54 min and was terminated under any of the following conditions: (1) core temperature reached 38.5 °C; (2) heart rate reached 95% of the age-predicted maximal HR (220-age) [29]; or (3) the subject requested to stop. Subsequently, an active recovery was taken, which lasted 6 min, before the subject took passive recovery. During the passive recovery, subjects sat in the chair for 30 min with the personal cooling vest. Following recovery, the subjects performed the 2nd intermittent running. A registered nurse was engaged inside the chamber to provide medical care in case of emergency throughout the experiment.

### 3.2. Measurements and Calculations

Subjects’ perception (thermal sensation) and physiological (core temperature, skin temperature, and heart rate) responses were measured in the experiments. Subjects were required to rate their thermal sensation (TS) according to the ASHRAE 7-point scale [30] every 6 min during rest, and 3 min during the other exercises and recovery. The rating of TS ranges from 1 (cold) to 7 (hot), and the rating scale is shown in Table 1. Core temperature (*T*_c_) was monitored by an ingestible capsule (CoreTemp, HQI, Boston, MA, USA) swallowed 4–6 h before tests and recorded by a CorTemp data logger at a sampling frequency of 30 s. Heart rate (HR) was measured by a heart rate belt (Polar T34 Transmitter, Polar Electro, Kempele, Finland) every 30 s.

The local skin temperatures were monitored by four thermistor sensors (Especmic, Osaka, Japan) taped at four sites, namely the chest, forearm, thigh, and calf. Then, the mean skin temperature was calculated by Equation (8) using the four local skin temperatures [31]:*T*_skin_ = 0.3(*T*_chest_ + *T*_forearm_) + 0.2(*T*_thigh_ + *T*_calf_),(8)
where *T*_chest_, *T*_forearm_, *T*_thigh_, and *T*_calf_ are the temperatures at the chest, forearm, thigh, and calf, respectively.

Physiological heat strain was rated according to a physiological strain index (PSI) proposed by Moran et al. [18,19], which can be calculated as the following:(9)$$\mathrm{PSI}=5\times \frac{T_{c}-T_{c0}}{39.5-T_{c0}}+5\times \frac{\mathrm{HR}-{\mathrm{HR}}_{0}}{180-{\mathrm{HR}}_{0}},$$
where *T*_c_ and HR are simultaneous measurements taken at any time; *T*_c0_ and HR_0_ are the minimum core temperature and heart rate, respectively.

## 4. Results

### 4.1. Physiological Responses

Heart rate, skin temperature, and core temperature are the main indicators for human physiological response under heat stress, and their mean time-dependent changes of all subjects are shown in Figure 3. As predicted, no difference is found between the heart rate plots of CON and PCV during stages 1–3 (pre-exercise rest, 1st run, and active recovery). The PCV plot begins to detach from the CON plot at the beginning of the passive recovery stage. Similar trends were also found in the plots of core temperature and skin temperature. It was noted that the values of heart rate, core temperature, and skin temperature during the passive recovery stage in PCV were significantly lower than those in CON, which means that PCV significantly reduces the heat strain. In addition, participants run for a longer time during the 2nd run. The longer duration of the 2nd run meant a longer work time and demonstrated the pre-cooling effects of PCV.

### 4.2. TS, PSI, Q, and CHS

Comparisons of TS, PSI, *Q*, and CHS dynamics in CON with those in PCV during the last two stages are shown in Figure 4, since they are the same during the first three stages, as aforementioned. TS and PSI are presented as the mean and standard deviation. TS in CON showed a slight decrease from 6.25 to 5.25 during the passive recovery and a subsequent rise with running up to 7, the maximum of TS (very hot). However, the value in PCV dropped more significantly from 4.33 to 2.83 during the passive recovery, and then rose sharply to about 6.2 during the 2nd run (see Figure 4a). The TS dynamic suggests that the PCV can significantly improve the subjective perception.

The trends of PSI in CON and PCV groups are similar (Figure 4b). They both drop slowly during passive recovery, and then rise sharply in the 2nd run. In addition, PSI shows an obvious periodic change with the periodic workload. The lower PSI in PCV reinforces the evidence that PSI has the ability to differentiate the strain level between clothing types [23].

Figure 4c,d show the mean *Q* and CHS. It can be seen in Figure 4c that *Q* has a different dynamic from TS or PSI. Firstly, *Q* does not drop during passive recovery, but rises with time in PCV. Secondly, *Q* during the second run is larger in PCV than that in CON, perhaps because of the larger change rates of *T*_sk_ and *T*_c_ in PCV. Unlike *Q*, CHS shows a similar dynamic to PSI (Figure 4b,d), which means CHS might be a better indicator for heat strain than *Q*.

## 5. Discussions

### 5.1. Relationships between TS with Q and CHS

As aforementioned, the relationship between thermal sensation and thermal load remains an arguable question. Some studies suggested that human thermal sensation was linearly associated with thermal load or the mean body temperature (heat storage), whereas some suggested it was associated with skin temperature and its change rate [32,33]. To examine the applicability of the thermal load or heat storage for thermal sensation assessment in hot-humid environments, their data are presented in Figure 5, and their relationships were explored.

It was found in Figure 5a that TS in CON group almost remained unchanged and just fluctuated near the value of 6, though *Q* varied from −50 to 150 W/m^2^. The linear trend between thermal load and thermal sensation was not obvious or convincing. A further correlation analysis between them shows that their linearity is indeed quite weak (*r*^2^ = 0.2209), though significant (*p* = 0.003). Thus, it can be derived that human thermal sensation was difficult to be well predicted by thermal load, at least, not well predicted in a linear way. Although the plot of TS versus CHS (Figure 5b) shows a stronger linear trend (*r*^2^ = 0.5041, *p* = 0.000), the increase of linearity and the reliability of predicting TS by *Q* remains limited. To evaluate the TS more precisely, a more complex model including *T*_sk_ and its change rate, as well as CHS or TS should be considered [34].

### 5.2. Relationship between PSI and CHS

As shown in Figure 4b,d, PSI and CHS showed a similar dynamic, so it was reasonably assumed that CHS can be a substitute for PSI for assessing the heat strain in some cases where skin temperature was easily monitored (such as by Infrared Thermometry [25]). The calculated PSI and CHS of subjects in CON and PCV groups during recovery were presented in Figure 6. Significant linear relationships were found between PSI and CHS for all four subjects and the two clothing types (PVC and no-PCV).

It was noted that the data of the two clothing types nearly plotted at a same line, which means that the development of PSI with CHS in PCV follows the same rule as that in CON. Regarding this, all the samples (*n* = 960) were pooled and plotted in Figure 7.

As shown in Figure 7, PSI is highly linear to CHS (*r*^2^ = 0.8407), shown by the following equation:PSI = 0.3 + 0.0265 CHS.(10)

PSI can be well predicted by CHS in Equation (10) whether subjects wear PCV or not. Thus, CHS also has the ability to differentiate the strain level between clothing types as PSI [31,32], and is reliable for assessing the physiological heat strain of still subjects in hot-humid environments. It should be noted that the data of PSI and CHS in Figure 7 were calculated using the *T*_c_, HR, or *T*_sk_ during the recovery stage. Thus, the application of CHS for heat strain assessment should be limited in rest condition. Further research will be carried out to develop a modified CHS which can assess heat strain during exercise.

## 6. Conclusions

Temperature based indices (thermal load and heat storage) were developed based on a dynamic two-node thermal balance model, and their applicability for assessing human thermal sensation and heat strain were examined. The main results are summarized as follows:

(1) The PCV can alleviate human heat strain, which is evident for subjects in PCV experiencing a lower heart rate, core temperature, and skin temperature.

(2) It is difficult to precisely predict human thermal sensation by a single factor model of thermal load or cumulative heat storage (CHS). A more comprehensive model including *T*_sk_ and its change rate, as well as thermal load or CHS, should be developed.

(3) The good linear relationship between PSI and CHS evidenced that CHS is an applicable and reliable index for human heat strain assessment during rest, and that it has the ability to differentiate the heat strain levels between different clothing types. Thus, the non-invasive diagnosis technique of IRT can be used in human heat strain monitoring through the temperature based index of CHS.

## Figures and Tables

**Figure 1 materials-13-01801-f001:**
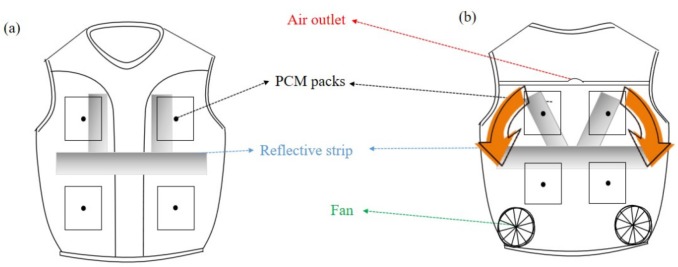
Illustration of the personal cooling vest (PCV): (**a**) front view; (**b**) back view.

**Figure 2 materials-13-01801-f002:**
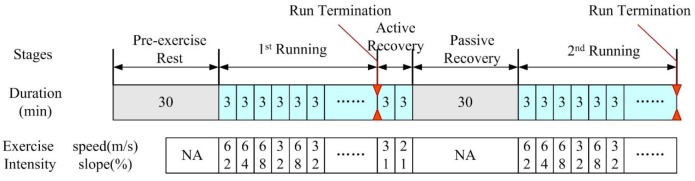
Protocol for the experiment. The 1st and 2nd run cycles terminate when *T*_c_ (temperature of the core body) = 38.5 °C, or HR (heart rate) = 0.95(220 − Age), or the subject requests to stop.

**Figure 3 materials-13-01801-f003:**
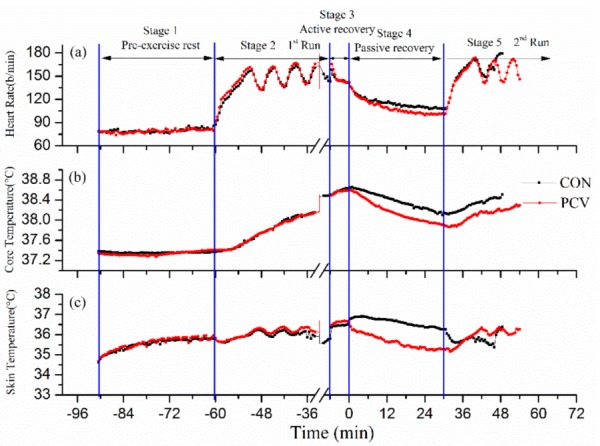
Comparisons of (**a**) heart rate, (**b**) core temperature, and (**c**) skin temperature in control group (CON) with those in PCV.

**Figure 4 materials-13-01801-f004:**
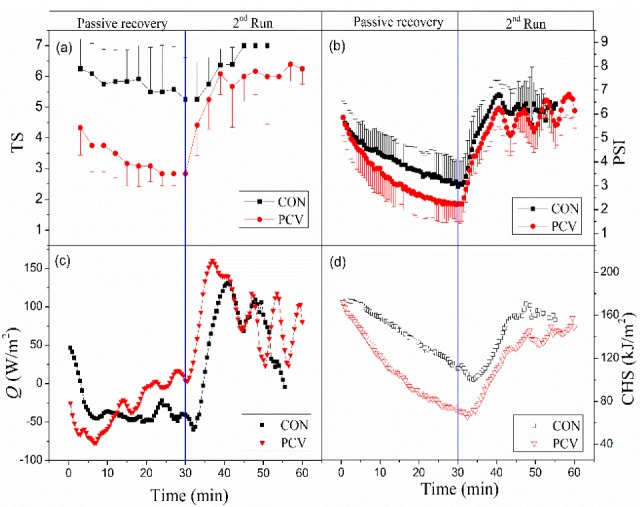
Comparisons of (**a**) thermal sensation (TS), (**b**) physiological strain index (PSI), (**c**) thermal load (*Q*), and (**d**) cumulative heat storage (CHS) in CON with those in PCV.

**Figure 5 materials-13-01801-f005:**
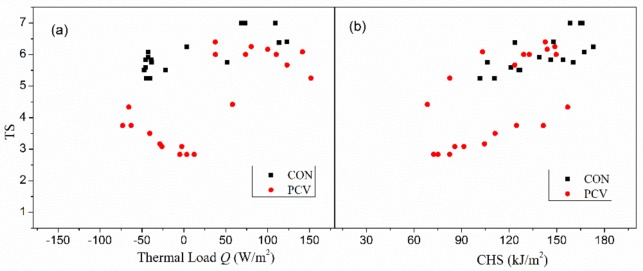
Relationships of thermal sensation with (**a**) *Q*, and (**b**) CHS.

**Figure 6 materials-13-01801-f006:**
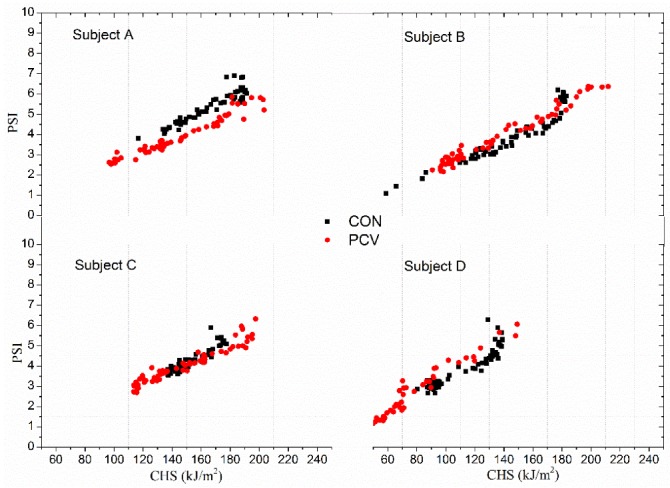
PSI versus CHS for subjects in CON and PCV.

**Figure 7 materials-13-01801-f007:**
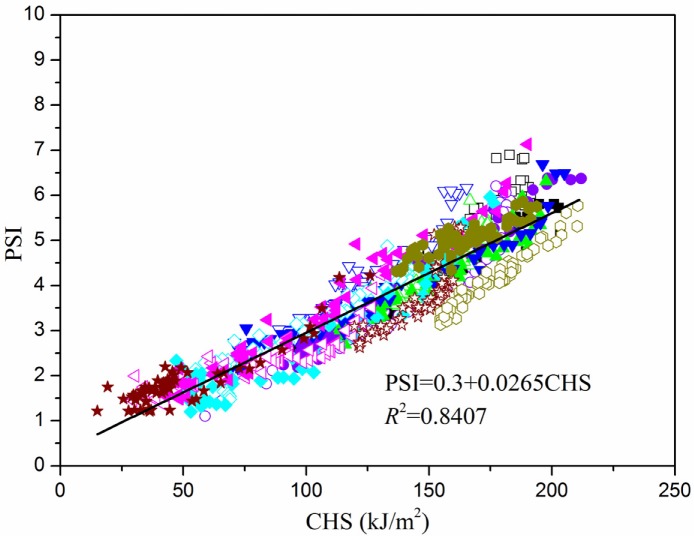
Relationship between PSI and CHS (*n* = 960).

**Table 1 materials-13-01801-t001:** Seven-point ASHRAE thermal sensation scale [30].

Scale	1	2	3	4	5	6	7
**TS**	Cold	Cool	Slightly cool	Neutral	Slightlywarm	Warm	Hot

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
