# Peer review of "Experimental Study on the Efficacy of a Novel Personal Cooling Vest Incorporated with Phase Change Materials and Fans"

_materials, 2020, doi:10.3390/ma13081801_

Round 1

Reviewer 1 Report

In my opinion, this is an interesting manuscript, describing the testing of a personal cooling vest, to assist manual workers (e.g. construction workers) in hot climates. The study was performed well and the manuscript well-written. I am happy to recommend publication subject to addressing a few minor issues.

L51: 'Two widely accepted heat strain index...' should be 'Two widely accepted heat strain indices...'

L55: '...could be monitored by the infrared thermography...' 'The' is wrong; it should be '...could be monitored by infrared thermography...'

L56: 'anal' should be 'rectum'.

L96-97: It would be better to list the parameters and their values in the same order.

L102: The experimental protocol is illustrated in Fig. 2 (not Fig. 1 as stated in the manuscript).

L108-110: The last sentence is unnecessary; it is effectively a repeat of what was stated at lines 84-89.

L127-128: Is rather confusing. It appears to suggest the 'active recovery' step was omitted. Can the authors clarify this in the manuscript, please.

L161: 'approved' is the wrong word. 'showed' or 'demonstrated' would be better words.

L197: 'and convinced' is the wrong expression. Do the authors mean 'or convincing'? Please clarify.

L200: 'verse' should be 'versus'.

Reviewer 2 Report

The paper presents very interesting results of the investigation of cooling performance of cooling vest with PCM. However, the paper requires refinement before publication.

Main comments:

The title Chapter 2 should better reflect its content. It is rather on mathematical models, methods (experiments) are described in Chapter 3.

Line 59: Q is heat flow rate, not heat flux

In line 64 there should be “latent”, not “sensible”

All variables should be defined (e.g. in Nomenclature)

The sentence in lines 108-110 is not necessary. It is a repetition of the information that was given earlier (84-89)

Line 199: predicted, not predict

Line 200: versus, not verse

There is a big mess in the literature list (Reference). It looks like it was just taken from the previous paper without any changes. It is supposed that there are also many incorrect references in the Introduction (state-of-the-art).
